# Persistence of Pneumococcal Carriage among Older Adults in the Community despite COVID-19 Mitigation Measures

Anne L. Wyllie,[a] Sidiya Mbodj,[a] Darani A. Thammavongsa,[a] Maikel S. Hislop,[a] Devyn Yolda-Carr,[a] Pari Waghela,[a] Maura Nakahata,[a] Anne E. Stahlfeld,[a] Noel J. Vega,[a] Anna York,[a] Orchid M. Allicock,[a] Geisa Wilkins,[b] Andrea Ouyang,[b] Laura Siqueiros,[b] Yvette Strong,[b] Kelly Anastasio,[b] Ronika Alexander-Parrish,[c] Adriano Arguedas,[c] Bradford D. Gessner,[c] Daniel M. Weinberger[a]

[a]Department of Epidemiology of Microbial Diseases, Yale School of Public Health, New Haven, Connecticut, USA
[b]Yale Center for Clinical Investigation, New Haven, Connecticut, USA
[c]Medical and Scientific Affairs, Pfizer Inc, Collegeville, Pennsylvania, USA

**ABSTRACT** Reported rates of invasive pneumococcal disease were markedly lower than normal during the 2020/2021 winter in the Northern Hemisphere, the first year after the start of the COVID-19 pandemic. However, little is known about rates of carriage of pneumococcus among adults during this period. Between October 2020-August 2021, couples in the Greater New Haven Area, USA, were enrolled if both individuals were aged 60 years and above and did not have any individuals under the age of 60 years living in the household. Saliva samples and questionnaires regarding social activities and contacts and medical history were obtained every 2 weeks for a period of 10 weeks. Following culture-enrichment, extracted DNA was tested using qPCR for pneumococcus-specific sequences *piaB* and *lytA*. Individuals were considered positive for pneumococcal carriage when Ct values for *piaB* were ≤40. Results. We collected 567 saliva samples from 95 individuals (47 household pairs and 1 singleton). Of those, 7.1% of samples tested positive for pneumococcus, representing 22/95 (23.2%) individuals and 16/48 (33.3%) households. Study participants attended few social events during this period. However, many participants continued to have regular contact with children. Individuals who had regular contact with preschool and school-aged children (i.e., 2 to 9 year olds) had a higher prevalence of carriage (15.9% versus 5.4%). Despite COVID-19-related disruptions, a large proportion of older adults continued to carry pneumococcus. Prevalence was particularly high among those who had contact with school-aged children, but carriage was not limited to this group.

**IMPORTANCE** Carriage of *Streptococcus pneumoniae* (pneumococcus) in the upper respiratory tract is considered a prerequisite to invasive pneumococcal disease. During the first year of the COVID-19 pandemic, markedly lower rates of invasive pneumococcal disease were reported worldwide. Despite this, by testing saliva samples with PCR, we found that older adults continued to carry pneumococcus at pre-pandemic levels. Importantly, this study was conducted during a period when transmission mitigation measures related to the COVID-19 pandemic were in place. However, our observations are in line with reports from Israel and Belgium where carriage was also found to persist in children. In line with this, we observed that carriage prevalence was particularly high among the older adults in our study who maintained contact with school-aged children.

**KEYWORDS** pneumococcus, saliva, surveillance, carriage, COVID-19 pandemic

Address correspondence to Anne L. Wyllie, anne.wyllie@yale.edu.

The authors declare a conflict of interest. A.L.W. has received consulting and/or advisory board fees from Pfizer, RADx, Diasorin, PPS Health, Co-Diagnostics, Filtration Group, and Global Diagnostic Systems for work unrelated to this project, and is Principal Investigator on research grants with Pfizer, Merck, Flambeau Diagnostics, Tempus Labs, and The Rockefeller Foundation to Yale University. D.M.W. has received consulting fees from Pfizer, Merck, GSK, Affinivax, and Matrivax for work unrelated to this project and is Principal Investigator on research grants and contracts with Pfizer and Merck to Yale University. This work has been previously presented in part at IDweek 2021 (virtually); the 15th European Meeting on the Molecular Biology of the Pneumococcus, Liverpool, United Kingdom; and the 12th International Symposium on Pneumococci and Pneumococcal Diseases (ISPPD-12), Toronto, Canada. Adriano Arguedas (A.A.) is a Pfizer employee and may have stock options at Pfizer.

**M**itigation measures that have been used to reduce the burden of COVID-19 have also had a profound effect on the incidence of disease caused by other pathogens. Major respiratory viruses, including influenza, respiratory syncytial virus, and human metapneumovirus, largely disappeared as causes of disease during the 2020-21 winter season in the Northern hemisphere (1–3). Invasive pneumococcal disease (IPD)

declined sharply in the spring of 2020 across all age groups and did not return to near regular levels until spring or summer of 2021 (2, 4–6).

It was initially assumed that the reduction in the incidence of IPD was due to reduced transmission of the bacteria resulting from the implementation of non-pharmaceutical interventions. Pneumococcus is commonly carried in the upper respiratory tract of young children, however, the prevalence of pneumococcal carriage in children was near normal levels during 2020 to 2021 (3, 7). This demonstrates that children were still being exposed to and acquiring pneumococcus but not getting sick. Therefore, the decline in IPD observed in children might instead be related to the absence of infection by common respiratory viruses, which are thought to increase the risk of severe pneumococcal disease (3). Additionally, some of the decline in IPD in children and adults could also be related to reduced health care seeking or changes in diagnostic practices during the pandemic (8).

While transmission of pneumococcus among children continued at high levels during the first year of the pandemic, social distancing and other mitigation measures could have reduced the amount of contact and transmission from children to adults. Children are a major source of exposure of pneumococcus for the adult population (9–11), which was most evident from the sharp drop in pneumococcal disease in adults that occurred following the introduction of pneumococcal conjugate vaccines in children (12). Therefore, we expected that reductions in contact between children and adults might have occurred during the COVID-19 pandemic, leading to some of the reduction in IPD observed in adult populations. We evaluated carriage rates among adults ≥60 years of age living in the community in the United States during 2020 to 21 who were participating in an ongoing longitudinal carriage study. Detailed information about their activities and contacts, along with their carriage status, provides important insights into the potential drivers of pneumococcal epidemiology in older adults.

## RESULTS

**Population characteristics.** From October 30th, 2020 through June 2021, 95 individuals from 48 households were sampled and completed all 6 visits. One household was composed of a single individual who was enrolled into the study due to residing in a living facility for older adults with high levels of contact between residents; due to the pandemic-related community restrictions in place, we were unable to enroll more individuals from the same facility. The mean age was 71 years (range 60 to 86) (Fig. S1). Of the 570 samples collected, 3 were not tested due to low collection volume ($n = 1$) or a weather-related delay (2 weeks) in transporting samples to the lab ($n = 2$). Of the study participants, 72% were white, and 42% had a bachelor's degree or higher. Eight individuals had a positive test for SARS-CoV-2 prior to enrollment in the study. None of the participants reported a positive test for SARS-CoV-2 while enrolled in the study nor tested positive for SARS-CoV-2 in any of the collected samples.

**Prevalence of pneumococcal carriage.** Overall, 40/567 (7.1%) of samples tested positive for pneumococcus based on *piaB*, with 22/95 (23.2%) individuals colonized on at least 1 time point (Fig. S2). Several individuals were colonized at multiple time points, including 2 individuals (participants 3 and 41) who were colonized throughout the 10 weeks sampling period and a third who was colonized at 5 of the 6 time points (participant 33). In 6/48 (12.5%) households, both members were carriers, though not necessarily at the same time point.

When samples were positive for both *piaB* and *lytA* ($n = 34$), there was good concordance in the bacterial density (Ct value) (Fig. S3). In some samples, the concentration of *lytA* was higher than *piaB* (lower Ct), likely reflecting the presence of non-pneumococcal *Streptococci* spp. in addition to pneumococcus (Fig. S3) (13, 14). In addition to higher specificity, the sensitivity of the *piaB* assay was also slightly higher, resulting in some samples near the limit of detection that were positive for *piaB* and negative for *lytA* (Fig. S3). It is for these reasons that when determining sample positivity, we relied on *piaB* alone.

To confirm the qPCR results, we used traditional culture-based methods, which are feasible when a high concentration of pneumococcus is detected (15). From the 40

**TABLE 1** Relationship between pneumococcal detection and activities outside the home during the previous 2 weeks

| Activities outside the home | $N^a$ | Pneumococcus (*piaB*+) | Percent positive |
|---|---|---|---|
| Activities with family | 74 | 10 | 13.5% |
| Activities with friends | 38 | 0 | 0% |
| Fitness activities | 13 | 0 | 0% |
| Activities at community centers | 8 | 0 | 0% |
| Other social activity | 115 | 5 | 4.3% |
| Total* | 579 | 40 | 6.9% |

[a]Numbers of visits at which people reported these activities. Participants were only asked about recent activities at visits 2 to 6.

samples that tested positive for *piaB*, 5 isolates, all from the same individual (participant 41), were successfully isolated and identified as pneumococcus. Each of the isolates was optochin sensitive, and the presence of both *piaB* and *lytA* genes were confirmed by qPCR. All isolates were identified as serotype 15B/C by latex agglutination.

**Reported activities outside of the home.** During this period, which included the first winter of the COVID-19 pandemic, participants continued to take part in some social activities outside the home (Table 1). The most commonly reported activities were gathering with family (40%) and friends (29%). Just 6% of participants reported taking part in activities at a community center, and 8% participated in fitness activities. The prevalence of pneumococcal carriage was modestly higher among those who reported activities with family (13.5%) compared with those participating in other social activities or those reporting no social activity (8.1%).

**Prevalence is higher among those in contact with children.** The prevalence of pneumococcal carriage was substantially higher among individuals who had contact with children (13.0% versus 3.5%, $P = 0.002$) (Fig. 1 and Table 2). Participants who reported recent contact with <5 year olds and 5 to 9 year olds had elevated prevalence (17.5%, $P = 0.001$; 15.9%, $P = 0.007$, respectively) (Table 2). Prevalence was not notably higher among those reporting contact with children older than 10 years of age (10.4%, $P = 0.47$). While the numbers are sparse, further subdividing the <5 year old population demonstrates progressively higher prevalence among those reporting contact with children <12m (11.9%), 12 to 23 months (13.3%) and 24 to 59 months (20.5%).

## DISCUSSION

Despite sharp declines in reported rates of IPD among adults during the first winter season of the COVID-19 pandemic (2020 to 21), older adults residing in the community continued to carry pneumococcus at levels consistent with what has been seen in other pre-pandemic studies of older adults that used similar molecular methods (16, 17). However, this study was conducted during a period when a reversion to stricter transmission mitigation measures was implemented throughout the state of Connecticut due to a resurgence in COVID-19 cases (18, 19). In the Greater New Haven Area, mask mandates were enforced in public spaces and had high adherence (20), and usual community activities were canceled throughout the study period. Study participants reported few activities outside of the home, even as restrictions were eased in March 2021 (21). However, many individuals did continue to have regular contact with their families, including young children, and these participants had particularly high rates of carriage. This is consistent with studies conducted among younger adults that children are a major driver of transmission of pneumococcus in the community (22, 23).

In this study setting, the period prevalence of pneumococcal carriage detected in this population of ≥60 year olds in the Greater New Haven Area was 23.2%. While other carriage studies conducted in older adults, also in the US, have reported lower rates of carriage (24), the difference in carriage rates can likely be attributed to sample and testing methodologies. As observed in the current study, saliva-based approaches and those that use qPCR following culture-enrichment tend to have higher sensitivity than those based on swabs and culture alone (15, 16). By including oropharyngeal swabs in addition to nasopharyngeal swabbing, Branche et al. reported a similar longitudinal carriage rate

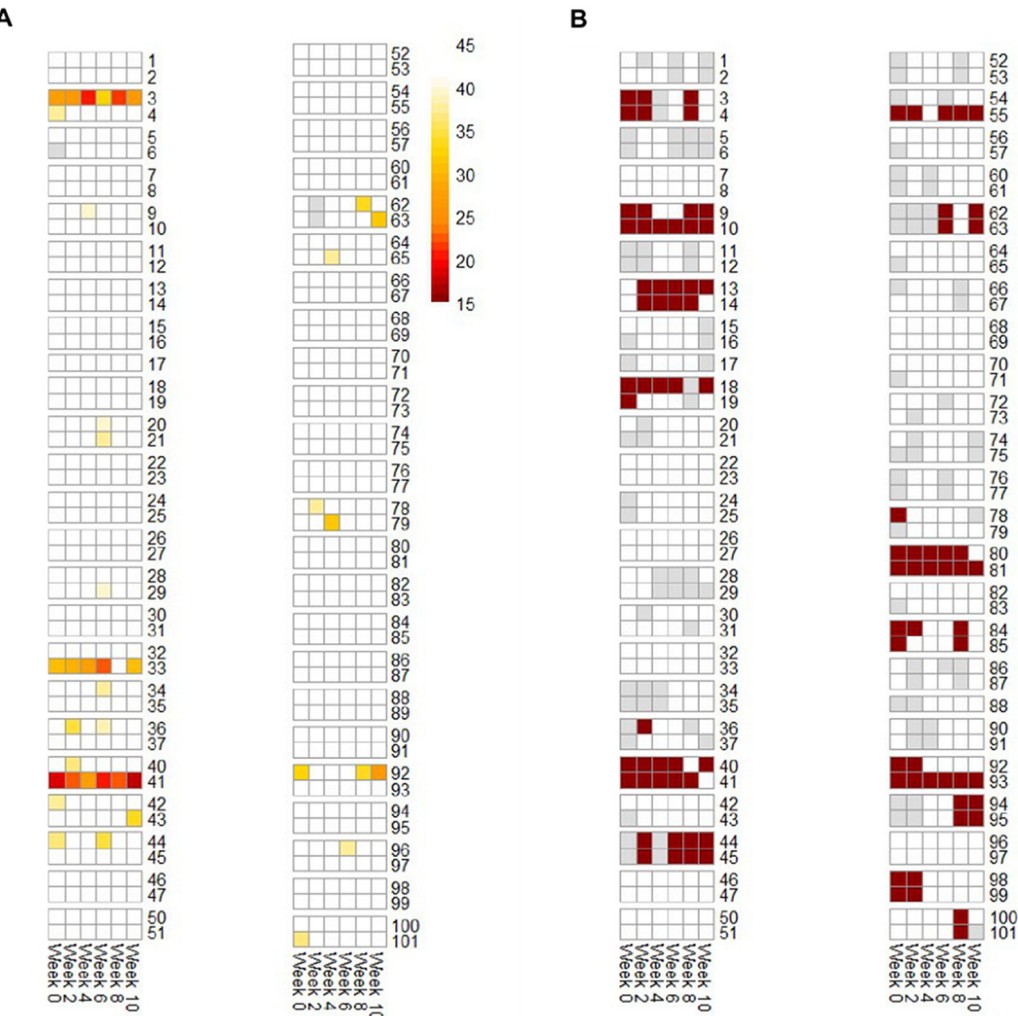

**FIG 1** (A) Detection of pneumococcus as measured by Ct values from PCR assays targeting pneumococcal gene, *piaB*. Colored boxes indicate the individual is colonized, with darker colors indicating a higher presence of pneumococcus (lower Ct values). (B) Contact with children. Maroon indicates the individual reported contact with children, white indicates no reported contacts, gray indicates the question was not answered or the survey is missing.

(≥25%), though did not detect an effect of contact with children (17). In their analyses however, Branche et al. considered contact with any children under 5 years of age (17) and as our analyses show, children under 2 years of age contribute little to transmission (25). This is also reflected in reports on their little contribution to disease incidence in older adults (25). When contact with children is not stratified by age group, and the majority of child contacts are younger, the effect of transmission is harder to detect. In this study, we asked study participants the ages of children with whom they interacted to enable these stratified analyses.

When surveying pneumococcal carriage using saliva samples, 1 limitation is the difficulty in isolating individual pneumococci due to the highly polymicrobial nature of saliva. Plating saliva on agar plates results in a solid lawn of diverse, bacterial growth. Even when heavily diluting the saliva prior to plating, pneumococcus typically comprises a minority. This greater abundance of growth compared to both nasopharyngeal and oropharyngeal swabs can make it near impossible to isolate individual colonies (22, 26). Thus, for the detection of pneumococcus we must rely on alternative methods. The method employed in the current study, testing DNA extracted from culture-enriched saliva in qPCR for pneumococcus genes *piaB* and *lytA* is one that has been used in previous studies (13, 16, 22, 27) and an approach that has demonstrated high

**TABLE 2** Relationship between pneumococcal detection and contact with children

| Contact with children | N | Pneumococcus (*piaB*+) | Percent positive |
|---|---|---|---|
| Contact with any children | | | |
| No contact | 318 | 11 | 3.5% |
| Contact | 192 | 25 | 13.0% |
| Missing | 69 | 4 | 5.8% |
| | | | |
| Contact <5 yr old children | | | |
| No contact | 443 | 23 | 5.2% |
| Contact | 80 | 14 | 17.5% |
| Missing | 56 | 3 | 5.4% |
| | | | |
| Contact 5 to 10 yr old children | | | |
| No contact | 441 | 24 | 5.4% |
| Contact | 82 | 13 | 15.9% |
| Missing | 56 | 3 | 5.4% |
| | | | |
| Contact >10 yr old children | | | |
| No contact | 456 | 30 | 6.6% |
| Contact | 67 | 7 | 10.4% |
| Missing | 56 | 3 | 5.4% |

sensitivity compared to other methods (28). However, we classified samples as positive for pneumococcus based on the detection of *piaB* alone, without a requirement for positivity of also *lytA* (29). While *lytA* has quickly become considered the gold standard gene target for the molecular detection of pneumococcus, there have been a growing number of reports of false positivity in this assay (30–32), predominantly due to the higher presence of closely related, non-pneumococcus *Streptococcus* spp. in oral samples which carry pneumococcal gene homologues (13, 14, 33). This was reflected in many samples that generated a large discrepancy in the Ct values reported for these two gene targets (Fig. S3). Additionally, despite numerous modifications, in our study setting, the sensitivity of the *lytA* qPCR assay was less than that of the *piaB* assay. This meant that there were instances toward the limit of detection of the qPCR assay where samples were positive for only *piaB*, which we confirmed with re-testing when the *piaB* Ct value was >35. Samples were only called positive if they tested positive twice or 2 out of 3 times in a tie-breaker situation. The limitation of relying upon only *piaB* is that *piaB* is missing from nontypeable pneumococci (34, 35) as well as in a few reported encapsulated strains (31). While our approach may have classified such samples as negative for pneumococcus, we expect these to rarely occur in this population of older adults, and if present, would mean we have a reported a conservative estimate of the true carriage rates.

In this first study season of an ongoing study investigating rates of pneumococcal carriage in Greater New Haven, we found little evidence of impact of pandemic mitigation measures on rates of the carriage of pneumococcus in older adults. During this period of reduced social contact, study results suggest school-aged children are the likely source of continued presence of pneumococcus in most study subjects. Importantly, the high rate of nonspecific signal detected in the widely used *lytA* qPCR assay demonstrates the importance of targeting multiple gene targets for reliable and specific detection of pneumococcus in oral samples (13). Follow-up studies with molecular serotyping of collected samples will provide greater insight into this observation and into transmission patterns of pneumococcus within households consisting of only individuals over the age of 60 years.

## MATERIALS AND METHODS

**Ethics.** This study was approved by the Institutional Review Board at Yale School of Medicine (protocol ID. #2000026100). Demographic data and samples were only collected after the study participant had acknowledged that they had understood the study protocol and provided verbal- or written-

informed consent. All participant information and samples were collected in association with anonymized study identifiers.

**Enrolment and eligibility.** These data are drawn from the first sampling year of an ongoing study of pneumococcal carriage. The broader study is designed to quantify and detect rates of acquisition of pneumococcal carriage among older adults and the role of household transmission between cohabitating older adults. We recruited pairs of individuals residing in the community in the greater New Haven area who were both 60 years of age or older and who did not have anyone under the age of 60 years living in the household. If an individual had symptoms of respiratory illness at time of consenting or had received antibiotics or pneumococcal vaccination within the past 4 weeks, the enrollment of that household pair into the study was delayed by up to 4 weeks. There were no exclusion criteria based on underlying health status.

**Sample and data collection.** Household pairs were sampled every 2 weeks, for a total of 6 visits covering 10 weeks. At each visit, the participants provided a self-collected saliva sample (36) into an empty 25 mL Eppendorf conical tube and answered questions about their social activities, doctors' visits, recent contact with children, and any respiratory symptoms they were experiencing or had experienced in the 2 weeks prior (Appendix). Study participants left their samples outside their front door for contactless-collection and transport back to the laboratory at room temperature.

**Sample processing and pneumococcal detection.** On arrival at the lab, 100 $\mu$L of raw saliva was plated on TSAII plates with 5% sheep's blood and 10% gentamicin and grown overnight at 37°C with 5% $CO_2$. Growth was harvested into 2100 $\mu$L of BHI supplemented with 10% glycerol and stored at −80°C until further processing. These samples were considered as culture-enriched for pneumococcus (16). DNA was extracted from 200 $\mu$L of each culture-enriched sample using the MagMAX Viral/Pathogen Nucleic Acid isolation kit (ThermoFisher Scientific) on the KingFisher Apex (ThermoFisher Scientific) with a modified protocol. Briefly, samples first underwent an extended digestion step, with 10 $\mu$L of proteinase K added to each sample and were incubated at 56°C for 10 min followed by heat inactivation of the proteinase K at 95°C for 10 min. Binding buffer and magnetic beads (25 $\mu$L) were added separately before proceeding with the KingFisher Apex extraction protocol which included an additional elution step, eluting extracted DNA into 2 plates of 50 $\mu$L of elution buffer. Purified DNA was tested by qPCR using primers and probes specific for 2 pneumococcal genes: *piaB* (15, 37) and *lytA* (38). The assays were carried out in 20 $\mu$L reaction volumes using SsoAdvanced Universal Probe Supermix (Bio-Rad, USA), 2.5 $\mu$L of genomic DNA and primer/probe mixes at concentrations of 250 nM (Iowa Black quenchers) for *piaB* (1 $\mu$L per reaction) and *lytA* (1.2 $\mu$L per reaction). DNA of *S. pneumoniae* serotype 19F was included as a positive control in every run. Assays were run on a CFX96 Touch (Bio-Rad) under the following conditions: 95°C for 3 min, followed by 45 cycles of 98°C for 15 s and 60°C for 30 s. Because many other *Streptococci* have the *lytA* gene, a sample was only considered to be positive for pneumococcus if it was positive for *piaB* (29). Samples were classified as positive with a *piaB* cycle threshold (Ct) value of <40 by RT-qPCR. However, DNA templates which generated a *piaB* Ct value between 35 and 40 were re-tested in qPCR. Only samples that tested positive twice (or twice out of 3 tests when a discrepant result occurred) were reported as a positive result.

**Strain isolation and serotyping.** Cultured-enriched saliva samples in which a higher concentration of pneumococcus was detected (<30 Ct by qPCR) were re-visited in an attempt to isolate pure pneumococcus. Samples were serially diluted 10-fold over a range of $10^{-1}$ to $10^{-5}$ in 1X PBS. The $10^{-5}$ and $10^{-6}$ serially diluted samples were plated (100 $\mu$L) onto plain blood agar plates. After overnight incubation, culture plates were visually screened for pneumococcal-like colonies. Each colony of pneumococcus-like morphology was streaked onto a plain blood agar plate and then inoculated into 50 $\mu$L of elution buffer (ThermoFisher Scientific) in a microcentrifuge tube. From each saliva sample, a total of 20 colonies were streaked onto one plain blood agar plate, with colonies pooled by 5 in each microcentrifuge tube. Pooled bacterial colonies were incubated for 10 min at 95°C on a heating block, then tested in qPCR for *piaB* and *lytA* to confirm the identity of the isolates. Streaked isolates from the pooled boilate samples that generated any signal <40 Ct for either *piaB* or *lytA* were individually tested for optochin susceptibility. Optochin susceptible colonies which tested positive for both *piaB* and *lytA* were then serotyped by latex agglutination (Statens Serum Institut) (39).

**Detection of SARS-CoV-2.** All saliva samples were also tested for the presence of SARS-CoV-2 virus RNA using the extraction-free SalivaDirect assay (40). Briefly, 50 $\mu$L of each sample was heated at 95°C for 5 min before being tested in RT-qPCR for SARS-CoV-2 (41).

**Statistical analysis.** Differences in the frequency of categorical outcomes were compared using Fisher's Exact test in the R Statistical Software v4.1.2.

## SUPPLEMENTAL MATERIAL

Supplemental material is available online only.

**SUPPLEMENTAL FILE 1**, PDF file, 1 MB.

## ACKNOWLEDGMENTS

We thank the study participants for their time and dedication to our study.

This work was supported by Pfizer. This study was performed as a collaborative research project between researchers at Yale School of Public Health and Pfizer. The study protocol was designed by the Yale researchers in consultation with Pfizer. The

decision to publish was made by the Yale researchers in consultation with Pfizer; all authors agree with the decision to publish the results of this study.

A.L.W., A.A., and D.M.W. conceived the study. A.L.W., R.A.-P., A.A., B.D.G., and D.M.W. designed the study protocol. A.L.W., S.M., Y.S., K.A., M.N., and D.M.W. managed the study. G.W., A.O., L.S., and Y.S. collected the data. S.M., D.A.T., M.S.H., D.Y.-C., P.W., N.J.V., A.Y., and O.M.A. were responsible for sample receipt, processing, and testing. A.L.W., D.A.T., M.S.H., M.N., A.E.S., and D.M.W. performed the analyses and interpreted the data. A.L.W., D.A.T., M.S.H., and D.M.W. drafted the manuscript. All authors amended and commented on the final manuscript.

A.L.W. has received consulting and/or advisory board fees from Pfizer, RADx, Diasorin, PPS Health, Co-Diagnostics, Filtration Group, and Global Diagnostic Systems for work unrelated to this project, and is a Principal Investigator on research grants with Pfizer, Merck, Flambeau Diagnostics, Tempus Labs, and The Rockefeller Foundation to Yale University.

D.M.W. has received consulting fees from Pfizer, Merck, GSK, Affinivax, and Matrivax for work unrelated to this project and is a Principal Investigator on research grants and contracts with Pfizer and Merck to Yale University. This work has been previously presented in part at IDweek 2021 (virtually); the 15th European Meeting on the Molecular Biology of the Pneumococcus, Liverpool, United Kingdom; and the 12th International Symposium on Pneumococci and Pneumococcal Diseases (ISPPD-12), Toronto, Canada.

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
