## [Reviewer comments · Microbiology Spectrum]

Microbiology Spectrum

Persistence of pneumococcal carriage among older adults in the community despite COVID-19 mitigation measures

Anne Wyllie, Sidiya Mbodj, Darani Thammavongsa, Maikel Hislop, Devyn Yolda-Carr, Pari Waghela, Maura Nakahata, Anne Stahlfeld, Noel Vega, Anna York, Orchid Allcock, Geisa Wilkins, Andrea Ouyang, Laura Siqueiros, Yvette Strong, Kelly Anastasio, Ronika Alexander-Parrish, Adriano Arguedas, Bradford Gessner, and Daniel Weinberger

Corresponding Author(s): Anne Wyllie, Yale School of Public Health

Review Timeline:

Submission Date:	November 27, 2022
Editorial Decision:	December 21, 2022
Revision Received:	February 13, 2023
Accepted:	March 20, 2023

Editor: Rosemary She

Reviewer(s): The reviewers have opted to remain anonymous.

Transaction Report:

DOI: <https://doi.org/10.1128/spectrum.04879-22>

December 21, 2022

Dr. Anne L. Wyllie
Yale School of Public Health
Epidemiology of Microbial Diseases
New Haven, CT 06510

Re: Spectrum04879-22 (Persistence of pneumococcal carriage among older adults in the community despite COVID-19 mitigation measures)

Dear Dr. Anne L. Wyllie:

Thank you for submitting your manuscript to Microbiology Spectrum. Your manuscript has been reviewed by two experts in the field and they have provided many mostly minor suggestions for improving the clarity of the manuscript. When submitting the revised version of your paper, please provide (1) point-by-point responses to the issues raised by the reviewers as file type "Response to Reviewers," not in your cover letter, and (2) a PDF file that indicates the changes from the original submission (by highlighting or underlining the changes) as file type "Marked Up Manuscript - For Review Only". Please use this link to submit your revised manuscript - we strongly recommend that you submit your paper within the next 60 days or reach out to me. Detailed instructions on submitting your revised paper are below.

Link Not Available

Sincerely,

Tulip Jhaveri

Journals Department
Reviewer comments:

Reviewer #1 (Comments for the Author):

This study investigates the rates of pneumococcal carriage among older adults during the 2020/2021 winter in the Northern Hemisphere (one year after COVID19). The study design is easy to follow and clearly explained.

I have the following concerns:

1. Methods - Inclusion criteria: "We recruited pairs of individuals residing in the community in the greater New Haven area who

were both 60 years of age or older and who did not have anyone under the age of 60 years living in the household". So why include a singleton in the analysis?

2. Methods and Supplementary figure 2 - "Because many other Streptococci have the *lytA* gene, a sample was only considered to be positive for pneumococcus if it was positive for *piaB*". Also, in the SF2 I have noticed that if a sample was *piaB* positive and *lytA* negative was considered pneumococcus positive. Why? I was expected pneumococcus positive samples to be *piaB* and *lytA* positive. Did these samples were confirmed pneumococcus positive by culture or any other method?

3. Methods - "Pooled bacterial colonies were incubated for 10 159 minutes at 95{degree sign}C on a heating block, then tested in qPCR for *piaB* and *lytA*." It is unclear to me why isolated pneumococcal colonies were again tested using qPCR for *piaB/lytA*.

4. Results- Prevalence of pneumococcus: Are all household pairs male-female? Did you consider to include sex as factor of pneumococcal prevalence and/or transmission within a household?

Reviewer #3 (Comments for the Author):

Within this study, the authors aimed at investigating the Persistence of pneumococcal carriage among older adults during the COVID-19 pandemic. They found that a large proportion of older adults continued to carry pneumococcus despite COVID-19 measures being in place. Saliva samples have been used and there are many reasons why this is a good idea in principle. In particular, the authors spent a lot of time in optimizing the sampling and the processing procedures (not only in this but also in other studies). In conclusion, this is important work but perhaps certain points need to be addressed

Major points:

A recent study investigated oropharyngeal samples to measure pneumococcal carriage in adults (PMID: 32727860). While I fully agree that OP and saliva samples (this study) are different, I am slightly concerned re the performance of *piaB* and *lytA*. The authors of the msphere paper wrote that: Using additional targets *piaB* improved qPCR specificity in OP (as compared to *lytA* alone), although the PPV (42 to 53%) was still poor (However, they used a microarray rather than cultures as golden standard). I am therefore worried if the cut off for saliva samples 'positive for pneumococcal carriage was set when Ct-values for *piaB* were {less than or equal to}40'. Especially, as the authors found that only 5 out of 40 samples were culture positive. In line 249, this is explained by 2 studies showing reduced sensitivity of cultures (as compared to qPCR) but I am not sure that this is fully convincing. I suggest discussing the findings of the msphere paper in the discussion.

Minor points:

Line 119: Why did the authors decide to use saliva samples? NP or OP samples seem to show more pneumococcal carriage, no (see table 1 of PMID: 32877517)?

Line 126ff: For future studies, using SP2020 could also be a good idea (PMID: 30824850)?

Line 129; Culture enrichment (used in this study): Is broth enrichment in line with the WHO guidelines (PMID: 35314867)? In general, what are the differences of this protocol as compared to the WHO guidelines?

Line 161: Optochin susceptible colonies were 'only' serotyped or also species confirmed (with e.g. MALD TOF MS)? Using MALDI TOF MS for species detection (and/or sequencing) could also be useful for samples <40ct.

Line 171: Should the statistical analyses address the fact that household pairs were included? Perhaps a regression analysis should be done which also includes other factors?

Line 201: The total number of positive samples was 40 (true?), but I think the number of samples with <30 Ct must be smaller, yes? How many? It is kind of visible in figure 1 but did not spot this in the text.

Staff Comments:

Preparing Revision Guidelines

Please return the manuscript within 60 days; if you cannot complete the modification within this time period, please contact me. If you do not wish to modify the manuscript and prefer to submit it to another journal, please notify me of your decision immediately so that the manuscript may be formally withdrawn from consideration by Microbiology Spectrum.

Reviewer #1 (Comments for the Author):

This study investigates the rates of pneumococcal carriage among older adults during the 2020/2021 winter in the Northern Hemisphere (one year after COVID19). The study design is easy to follow and clearly explained. I have the following concerns:

1. Methods - Inclusion criteria: "We recruited pairs of individuals residing in the community in the greater New Haven area who were both 60 years of age or older and who did not have anyone under the age of 60 years living in the household". So why include a singleton in the analysis?

Response: *The singleton was enrolled early in the study. We had initially planned to enrol individuals living in assisted living facilities, including singletons in these settings due to their high contact patterns with others over the age of 60 years. The singleton included in this study was an individual living in an assisted living facility who reported high contact with others of 60 years but little contact with those under 60 years. We therefore proceeded with their enrolment but due to the community restrictions in place due to the pandemic, we were unable to engage with others in the facility, or other such facilities in the community and thus were unable to enrol others in such settings. We have clarified this in the Results section under 'Population characteristics'.*

2. Methods and Supplementary figure 2 - "Because many other Streptococci have the *lytA* gene, a sample was only considered to be positive for pneumococcus if it was positive for *piaB*". Also, in the SF2 I have noticed that if a sample was *piaB* positive and *lytA* negative was considered pneumococcus positive. Why? I was expected pneumococcus positive samples to be *piaB* and *lytA* positive. Did these samples were confirmed pneumococcus positive by culture or any other method?

Response: *In previous studies, *piaB* has been shown to be highly specific for pneumococcus. When implementing the PCR assays in our research lab, we found the chemistry of the *piaB* assay to be more sensitive than the *lytA* assay, even following multiple assay modifications. This can be seen in **Supplementary Figure 3**, where there are a disproportionate number of *piaB*+/*lytA*- samples close to the limit of detection. This has resulted in a small discrepancy between the expected *piaB* and *lytA* Ct values for a pneumococcus strain. This means that towards the lower limit of detection, samples may test positive for *piaB* while the *lytA* PCR assay is not sensitive enough to produce a concordant positive value and instead appears to be negative. When pneumococcus is detected towards this lower limit of detection by PCR, the amount of pneumococcus is a tiny fraction of the total bacteria in the sample, making isolation by culture impossible. To improve the reliability of our PCR results when we could not confirm by culture, when a sample tested >35 Ct for *piaB*, the DNA template was retested. Only samples that tested positive twice (or two out of three tests should a discrepant result occur) were included as a 'positive' result. We have updated the text in the Methods under 'Sample processing and pneumococcal detection' to clarify this further.*

3. Methods - "Pooled bacterial colonies were incubated for 10 minutes at 95°C on a heating block, then tested in qPCR for *piaB* and *lytA*." It is unclear to me why isolated pneumococcal colonies were again tested using qPCR for *piaB*/*lytA*.

Response: *We included this step as an additional confirmation of the culture-based approach. Despite positivity for optochin, we wanted to be certain that the colony isolated generated the typical signal expected from pneumococcus in the *piaB* and *lytA* PCR assays. We have clarified this in the Methods under 'Strain isolation and serotyping'.*

4. Results- Prevalence of pneumococcus: Are all household pairs male-female? Did you consider to include sex as factor of pneumococcal prevalence and/or transmission within a household?

Response: Household pairs were predominantly male-female but we had two households that were a female pair. We evaluated pneumococcal prevalence in relation to sex but found no difference between males and females, so we did not explore this further.

Reviewer #2 (Comments for the Author):

The article of Wyllie et al investigate the persistence of pneumococcal carriage among older adults in the community despite COVID-19 mitigation measures. The work has several strengths and a few aspects that should be improved.

ABSTRACT

"Individuals were considered positive for pneumococcal carriage when Ct-values for were ≤ 40 ."

Q: Please, specify criteria for *lytA*.

Response: While in previous studies we have generally looked for concordance between *lytA* and *piaB* Ct values when determining sample positivity, when implementing the assays in lab, even following repeated study modifications, the sensitivity of the *lytA* assay remained lower than that of *piaB*. This means that in samples with low bacterial load, a sample may be weakly positive for *piaB* but negative for *lytA*. Due to the specificity of *piaB* that we have observed for pneumococcal detection in the past, combined with re-testing of weakly positive samples and requiring 2 out of 3 tests to be positive for *piaB*, we made the decision to call positivity on *piaB* alone. This approach has also been applied in carriage studies in the Netherlands due to the increasing reports of false positivity in the *lytA* assay (<https://academic.oup.com/cid/article/73/9/e2680/5943453>). **We have clarified our approach in the text in the Methods under 'Sample processing and pneumococcal detection' and hope that this satisfies the reviewer.**

"couples in the Greater New Haven Area, USA, were enrolled if both individuals were aged 60 years and above and did not have any individuals under the age of 60 years living in the household." ... "We collected 567 saliva samples from 95 individuals (47 household pairs and one singleton)".

Q: Patients selection and results are discordant. Please clarify if patients selection includes only couples or singleton are also included.

Response: We thank the reviewer for raising this point - Reviewer 1 commented on the same. We hope that you find our response to Reviewer 1 and **our additions to the text in the Results section under 'Population characteristics'** acceptable for addressing this point.

IMPORTANCE

Q: replace "Streptococcus pneumoniae" by "Streptococcus pneumoniae" in cursive

Response: We have made sure that *Streptococcus pneumoniae* is formatted correctly in this section

INTRODUCTION

Please describe the temporal evolution of COVID-19 restrictions in Connecticut. When were these restrictions enforced and when did relaxation measures start?.

Response: *We thank the reviewer for this suggestion. We have updated the text accordingly.*

Please include invasive pneumococcal disease evolution in the same period and geographical area as those considered in this carriage study.

Response: *We agree with the reviewer that this would be an interesting point to include. However, the data on IPD trends in the local area are not yet publicly available so we are unfortunately not able to include this in our manuscript. Over the course of 2020, rates reflected worldwide trends however, with a disappearance of IPD reported. It is likely that cases of IPD started to increase towards the end of 2021 as other respiratory viruses began circulating again, yet our period of sampling concluded prior to this.*

METHODS

Please specify the time since samples collection to sample receipt in the laboratory (mean/median, minimum and maximum time). Discuss in limitation if results can vary if the sample arrive in the lab after 24 hours. How long were samples stored at room temperature before bacterial culture was performed? This data are crucial for bacterial culture sensitivity.

Response: *Samples were collected by the study participants either the night before, or morning of their contactless collection from the household. If self-collected the night before we asked participants to store their sample in their refrigerator until leaving it out for their scheduled collection time. While we received the time of sample collection from some study participants this was not consistent and many data points for this are missing. We had to adjust our study protocol due to pandemic restrictions and shifted from in-person to remote visits. To be certain that the detection of pneumococcus would remain stable in saliva samples, we conducted a number of stability experiments, which we report in a recent publication:*

<https://journals.asm.org/doi/10.1128/msphere.00331-22>. This shows that detection remains stable for up to 72 hours either in cold conditions or up to 30°C, thus encompassing our 12-16 hours max time that they may have been delayed):

"Because many other Streptococci have the *lytA* gene, a sample was only considered to be positive for pneumococcus if it was positive for *piaB*"

Q: *lytA* is a highly specific gene for pneumococcal detection in nasopharynx (See <https://www.ncbi.nlm.nih.gov/pmc/articles/PMC5627049/>) and it is the recommended gene by CDC (<https://www.cdc.gov/streplab/pneumococcus/resources.html>) for detection of *S.pneumoniae* by PCR. It is true that false positive results have been reported using both *LytA* gene and *PiAB* in saliva. <https://pubmed.ncbi.nlm.nih.gov/32727860/>). It could be interesting to include a table comparing rates of carriage according to bacterial culture, *lytA*, *piaB* and both targets together. Differences should be explained: e. g., which factors could be related with extremely low sensitivity of bacterial culture?. Which factors could be related with potential false positive by PCR?

Response: *We very much appreciate the importance that the reviewer sees in this. We agree with the reviewer and have been looking into this over the past several years. We recently reported on the detection of 'pneumococcus-specific' gene targets used in common PCR assay in both pneumococcus and non-pneumococcal Streptococcus spp. (PMCID: PMC5627049, <https://doi.org/10.1101/2022.11.20.22282557>). From this, it is clear that the discrepancy observed in many samples between the signals from the *lytA* and *piaB* PCR assays is being driven by the*

presence of *lytA* in non-pneumococcus *Streptococcus* spp, producing a resulting 'false-positive' for pneumococcus in these samples. This has also been independently reported by others (<https://www.sciencedirect.com/science/article/pii/S0732889316300724?via%3Dihub>). **Additional citations have also been added to the text.** It was based on this that another carriage study in the Netherlands also recently employed this approach (<https://academic.oup.com/cid/article/73/9/e2680/5943453>).

The low sensitivity of bacterial culture is simply due to the highly polymicrobial nature of saliva, which produces a solid lawn of growth on culture plates (as compared to nice, individual colonies from nasopharyngeal swabs). The greater abundance of other bacteria makes it harder to find by eye, the pneumococcal colony responsible for the *piaB* signal. **We have included this in the discussion.**

lytA is a virulence factor of pneumococcus. Discussion should be improved. What is the implication of *lytA* detection in oral streptococci (Pneumo or non-pneumo...)?

Response: The qPCR assays used here only targets a small region of *lytA* (~150bp) so it is not possible for us to infer from this detection, how homologous the entire gene sequence is to the full *lytA* gene in pneumococci. A study from Portugal however, looked into this (<https://www.sciencedirect.com/science/article/pii/S0732889316300724?via%3Dihub>) and found two non-pneumococcal streptococcal isolates that harbored a sequence homologous to pneumococcus. There is discussion in the literature as to whether the presence in *lytA* in non-pneumococcal streptococci is a result of divergent evolution from a common ancestor or as a result of horizontal gene transfer. As these typically commensal strains are seldom investigated, and since *lytA* is only one of many pneumococcal virulence factors, it is not possible to determine the implication of *lytA* detection in oral streptococci, especially from the limited gene segment that we're targeting, other than for confounding molecular assays for pneumococcal detection.

Statistical analysis of carriage rates should be performed with *piAB*, *lytA* and both targets.

Please include statistical analysis in tables 1 and 2.

Response: Due to the qPCR assay targeting *lytA* having a lower limit of detection as compared to that of *piAB*, combined with the high degree of confounding of the *lytA* assay by non-pneumococcal streptococci, conducting statistical analyses on *lytA* alone would not accurately reflect the true abundance of pneumococcus in this population.

Reviewer #3 (Comments for the Author):

Within this study, the authors aimed at investigating the persistence of pneumococcal carriage among older adults during the COVID-19 pandemic. They found that a large proportion of older adults continued to carry pneumococcus despite COVID-19 measures being in place. Saliva samples have been used and there are many reasons why this is a good idea in principle. In particular, the authors spent a lot of time in optimizing the sampling and the processing procedures (not only in this but also in other studies). In conclusion, this is important work but perhaps certain points need to be addressed

Major points:

A recent study investigated oropharyngeal samples to measure pneumococcal carriage in adults (PMID: 32727860). While I fully agree that OP and saliva samples (this study) are different, I am

slightly concerned re the performance of piaB and lytA. The authors of the msphere paper wrote that: Using additional targets piaB improved qPCR specificity in OP (as compared to lytA alone), although the PPV (42 to 53%) was still poor (However, they used a microarray rather than cultures as golden standard). I am therefore worried if the cut off for saliva samples 'positive for pneumococcal carriage was set when Ct-values for piaB were {less than or equal to}40'. Especially, as the authors found that only 5 out of 40 samples were culture positive. In line 249, this is explained by 2 studies showing reduced sensitivity of cultures (as compared to qPCR) but I am not sure that this is fully convincing. I suggest discussing the findings of the msphere paper in the discussion.

Response: *We thank the reviewer for this important discussion point. As the reviewer recognises, the referenced paper relied on microarray detection. In previous studies we, and others, have observed a high concordance between lytA and piaB when testing less microbially dense samples (ie, nasopharyngeal swabs) in qPCR. We have also observed in thousands of samples a good concordance between lytA and piaB Ct values, when there is no evidence of confounding non-pneumococcal streptococci. Despite the high rates of lytA in non-pneumococcal streptococci (<https://www.medrxiv.org/content/10.1101/2022.11.20.22282557v1>), piaB is still considered highly specific for encapsulated piaB strains; it is not often found in nontypeable strains (PMCID: PMC5627049). While there have been some reports that the piaB gene is not present in all pneumococcal serotypes, it is only a very few, strains that have been reported - it is not consistent to overall serotypes but limited to isolated strains. Thus, the most common pneumococcus strain that will be missed when relying on detection of piaB, alone, is non-typeables. While this can lead to an underestimation of true carriage prevalence, these non-typeable strains are also much less commonly detected in older adults.*

Minor points:

Line 119: Why did the authors decide to use saliva samples? NP or OP samples seem to show more pneumococcal carriage, no (see table 1 of PMID: 32877517)?

Response: *Few studies have compared the detection of NP, OP and saliva. While PMID: 32877517 is one of them, in other populations, members of this study team have previously found saliva to outperform other sample types, especially in older adults (PMCID: PMC4366201, PMCID: PMC5054371). Recognizing the gap in the literature, directly comparing saliva, OP and nasal wash samples, we initially planned to collect all three sample types from study participants. However, since OP and nasal wash samples (as well as NP samples) are invasive and/or specialized sample types, they require a trained healthcare professional to collect. As this study was conducted during the northern hemisphere autumn/winter season of 2020/2021, and was focused on the sampling of older adults, our environmental safety department did not approve the samples in which healthcare workers were required to unnecessarily interact with vulnerable members of the community due to the potential risk for transmission of SARS-CoV-2 between healthcare workers and study participants. Therefore, saliva self-collection provided a reliable solution for exploring carriage in this at-risk age group during the COVID-19 pandemic.*

Line 126: For future studies, using SP2020 could also be a good idea (PMID: 30824850)?

Response: *We thank the reviewer for this suggestion and agree with this. We are currently exploring the specificity of SP2020 in our collected saliva samples. From the samples tested so far, we see as much noise from SP2020 as with lytA, ie, there is no stronger concordance between lytA and SP2020 nor piaB and SP2020 as shown in PMID: 30824850. We are continuing to test many of our saliva samples to establish how this performs between different populations and age*

groups. In context of the current study however, it did not improve upon *piaB* detection but we would like to test it further before reporting on its performance in our community cohorts.

Line 129; Culture enrichment (used in this study): Is broth enrichment in line with the WHO guidelines (PMID: 35314867)? In general, what are the differences of this protocol as compared to the WHO guidelines?

Response: *The WHO guidelines were set forward in 2013. We have learned a lot since then. Those WHO guidelines still recommend nasopharyngeal swabs as the gold standard sample type and testing them with culture-based methods. They also recommend that when possible, an OP sample can also be collected and tested. We and others have demonstrated that qPCR detection improves detection in oral samples, and saliva can increase this further.*

In regards to broth enrichment, the WHO guidelines, as referred to in PMID: 35314867 state that more research into this is required. In the current study, we use culture-enrichment which is different to broth enrichment. We did conduct a small (non-published) study in 2014, comparing broth- and culture-enrichment and found somewhat comparable detection between the two approaches. However, we preferred the process of culture-enrichment when dealing with large numbers of samples. What we have previously shown however, is that culture-enrichment improves pneumococcal carriage detection - especially in oral samples where the overall fraction of pneumococcus is smaller in these polymicrobial samples as compared to the less dense nasopharyngeal samples (PMID: 25013895, figure 2).

Additionally, in the 2015 paper by Satzke et al., comparing serotyping methods across different centers, their study found the combination of culture-enrichment and qPCR to be one of the most sensitive methods for detection (m21; together with microarray, m4). Broth-enrichment, in place of culture-enrichment, did not perform as well (m20) -

<https://journals.plos.org/plosmedicine/article?id=10.1371/journal.pmed.1001903>.

Line 161: Optochin susceptible colonies were 'only' serotyped or also species confirmed (with e.g. MALD TOF MS)? Using MALDI TOF MS for species detection (and/or sequencing) could also be useful for samples <40ct.

Response: *Isolated colonies were confirmed with *lytA* and *piaB* qPCR on the basis that *piaB* has proved highly specific for pneumococcus. When both *lytA* and *piaB* were detected in an optochin susceptible colony, this confirmed to us the presence of pneumococcus. Unfortunately, due to the high presence of non-pneumococcus *Streptococcus* spp. in saliva samples, other methods, such as MALDI-TOF MS and sequencing are subject to confounding due to the high prevalence of homologous genes in these highly related species.*

Line 171: Should the statistical analyses address the fact that household pairs were included? Perhaps a regression analysis should be done which also includes other factors?

Response: *We thank the reviewer for this question. Since the observations from the household pairs are not fully independent, resulting P-values from this might be liberal. Adjusting for the correlation structure would require using a generalized estimating equation (GEE) framework, but the sample size here is too small to do this robustly. We are continuing to collect and analyze samples from subsequent seasons that allow for a fully adjusted longitudinal analysis in a Markov modeling framework, and the conclusions do not meaningfully change when doing this.*

Line 201: The total number of positive samples was 40 (true?), but I think the number of samples with <30 Ct must be smaller, yes? How many? It is kind of visible in figure 1 but did not spot this in the text.

Response: *14 samples produced piaB Ct values <30. However, any sample producing a piaB value >35 was re-tested to confirm detection. Samples had to test positive twice for piaB with Ct values <40 Ct (ie, consistent results) to be included as a 'positive' sample. **We have updated the text in the Methods to clarify this.***

March 20, 2023

Dr. Anne L. Wyllie
Yale School of Public Health
Epidemiology of Microbial Diseases
New Haven, CT 06510

Re: Spectrum04879-22R1 (Persistence of pneumococcal carriage among older adults in the community despite COVID-19 mitigation measures)

Dear Dr. Anne L. Wyllie:

Your manuscript has been accepted, and I am forwarding it to the ASM Journals Department for publication. You will be notified when your proofs are ready to be viewed.

Sincerely,

Rosemary She
Editor, Microbiology Spectrum
